# Recent Advances and Challenges in Halide Perovskite Crystals in Optoelectronic Devices from Solar Cells to Other Applications

**Seunghyun Rhee** [1] , **Kunsik An** [2,*] and **Kyung-Tae Kang** [2,*]

1  SKKU Advanced Institute of Nanotechnology, Sungkyunkwan University, Suwon 16419, Korea;
   rheejoe2@gmail.com
2  Manufacturing Process Platform R&D Department, Korea Institute of Industrial Technology (KITECH),
   Ansan 15588, Korea
*  Correspondence: kunsik1214@kitech.re.kr (K.A.); ktkang@kitech.re.kr (K.-T.K.); Tel.: +82-31-8040-6427 (K.A.);
   +82-31-8040-6428 (K.-T.K.)

**Abstract:** Organic-inorganic hybrid perovskite materials have attracted tremendous attention as a key material in various optoelectronic devices. Distinctive optoelectronic properties, such as a tunable energy band position, long carrier diffusion lengths, and high charge carrier mobility, have allowed rapid progress in various perovskite-based optoelectronic devices (solar cells, photodetectors, light emitting diodes (LEDs), and lasers). Interestingly, the developments of each field are based on different characteristics of perovskite materials which are suitable for their own applications. In this review, we provide the fundamental properties of perovskite materials and categorize the usages in various optoelectronic applications. In addition, the prerequisite factors for those applications are suggested to understand the recent progress of perovskite-based optoelectronic devices and the challenges that need to be solved for commercialization.

**Keywords:** peorvksite; perovskite solar cell; organic-inorganic halide structure; perovskite light emitting diode; perovskite photodetector



## 1. Introduction

With the expeditious growth of economic development and the demand for optoelectronic devices, organic-inorganic hybrid perovskites have attracted a great amount of attention as key materials in various fields, such as solar cells, light-emitting diodes [1–3], laser [4] and photodetectors [5,6]. By exploiting their superior properties, perovskite-based optoelectronic materials have developed at rapid speed [7–10]. Since the first demonstration was reported for perovskite solar cells (PSCs) in 2009 [11], the power conversion efficiency (PCE) of PSCs has improved considerably reaching ~25.5% within the past several years [12]. Moreover, the external quantum efficiency (EQE) of perovskite-based light-emitting diodes (PeLEDs) has now reached ~20% [13], which is comparable to that of organic light-emitting diodes (OLEDs) or quantum dot light-emitting diodes (QD-LEDs). Perovskite-based optoelectronic applications are now expanding and accomplishing astonishing performance in every field (Figure 1b–e).

Perovskite material has an inorganic and organic hybrid structure with the main formula $ABX_3$. The "A" and "B" denote two cations with different sizes, and "X" is an anion (Figure 1a). The common representative of these cations and anion is A = $Cs^+$, $CH_3NH_3^+$ ($MA^+$), $CH(NH_2)_2^+$ ($FA^+$), $CH_3CH_2NH_3^+$, B = $Pb^{2+}$, $Sn^{2+}$, $Cu^{2+}$, and X = $Cl^-$, $Br^-$, and $I^-$ (halogens). Based on the various combinations of A, B, and X, the perovskite material shows unique properties: a tunable band gap energy ranging from 1.6 eV ($FAPbI_3$) to 3.1 eV ($CsPbCl_3$) [14,15], long hole–electron diffusion length (>1 μm) [16,17], high charge carrier mobility [15,18], high photoluminescence quantum yields (PLQYs) [19,20] and low electronic trap states [21,22]. Thus, with such desirable features as light weight, flexible

form, low cost, and solution-based processability, perovskite-based optoelectronic devices are leading in various fields.

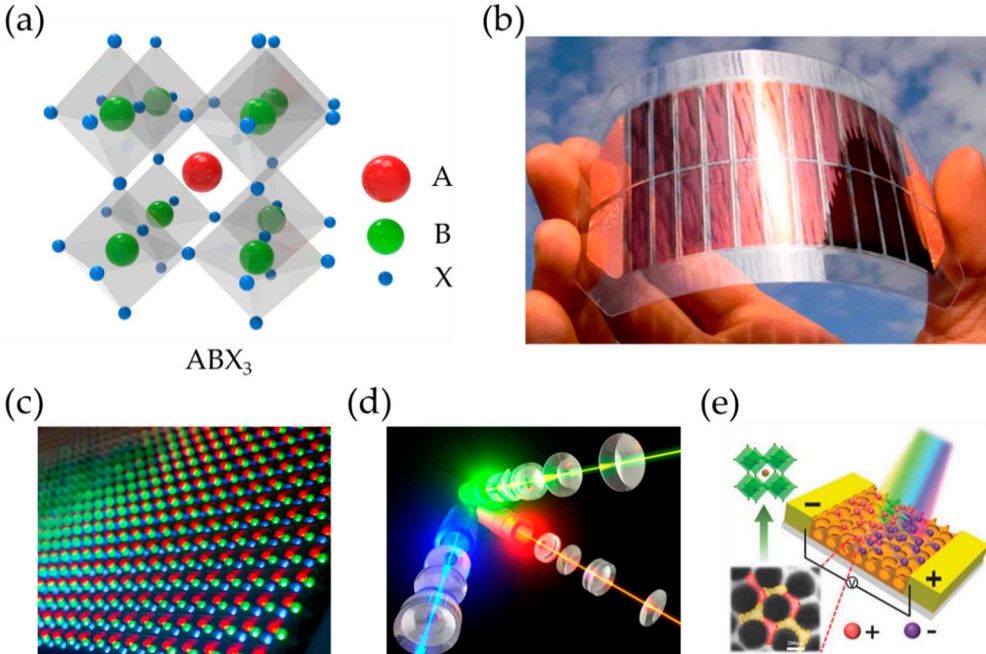

**Figure 1.** (**a**) Schematic diagram of crystal structure of hybrid perovskites and various optoelectronic applications, (**b**) solar cells, (**c**) LEDs, (**d**) lasers, and (**d**) photodetectors of perovskites. (**e**) Adapted with permission from [23].

Among various applications, PSCs are most widely being investigated. Most notably, PSCs with a PCE of 25.5% were produced by the Seok et al., which is comparable to the efficiency of commercialized traditional solar cells based on crystalline silicon and other combinations of inorganic semiconductors. In addition, benefitting from the near unity PL QY of perovskite nanocrystals, the EQE of PeLEDs has now reached 20% [13,24]. Since perovskite as emitting layer possesses a high degree of color purity for visible to infrared color ranges [25,26], PeLEDs promise the realization of full color displays with a wide color gamut exceeding the BT2020 standard. In laser and photodetector applications, perovskite also attracts a great deal attention due to its unique optoelectronic properties [27,28].

Halide perovskite crystals require different characteristics according to their applications. For example, in solar cells, low exciton binding energy is needed for efficient generation of electron-hole pairs, while LEDs require a highly radiative recombination process. The perovskite film properties such as layer thickness, surface characteristics, diffusion length, and trap state density largely differ from the chemical structure and film fabrication process. To identify the key issues in each perovskite-based application, we investigated the prerequisite factors in each application based on the representative studies. In this review, we briefly introduce the fundamental structure and characteristics of a halide-based perovskite layer and then highlight the key factors for achieving high performance in each application: solar cells, light-emitting diodes, lasers, and photodetectors. This is followed by a description of the remaining challenges. This review confirms the high potential of perovskite as a universal material solution for a wide range of optoelectronic applications, and enhances the understanding of key properties in each application for their practical commercialization.

## 2. Basic Structure and Optoelectronic Characteristics of Perovskite Crystals

### 2.1. Crystal Structure of Perovskite

The three-dimensional (3D) halide perovskite crystals have a common chemical formula $ABX_3$, where A and B denote cations and X is an anion. The ideal structure of

perovskite is a cubic structure, where $BX_6$ is located at the corners of the octahedral coordination surrounding the A cations in the center (Figure 1a) [29]. Although these cations and halide anion have multiple combinations of candidates, the Goldschmidt tolerance factor and the octahedral factor are necessary to for a stable $ABX_3$ structure [30,31]. The two factors are expressed as

$$\text{Tolerance factor}(t) = \frac{r_A + r_X}{\sqrt{2}(r_B + r_X)} \tag{1}$$

$$\text{Octahedral factor}(\mu) = \frac{r_B}{r_X} \tag{2}$$

where $r_A$ and $r_B$ are the ionic radius of the A and B cations and $r_X$ is the ionic radius of the anion. For a cubic perovskite structure, the tolerance factor t should be $0.9 \leq t \leq 1$, where t = 1 indicates perfect symmetry. When the t is $0.7 \leq t \leq 0.9$, the perovskite forms an orthorhombic or rhombohedral structure. Therefore, to synthesize a stable structure of perovskite crystals, the tolerance factor needs to be located between 0.7 to 1.0. Meanwhile, the octahedral factor μ must exceed 0.41 to form a stable octahedral coordination [32].

### 2.2. Optoelectronic Characteristics of Perovskite

2.2.1. Optical Characteristics of Perovskite: Absorption, Emission, and PL QYs

Perovskite as a light absorber (e.g., solar cells and photodetector) possesses strong optical absorption properties ranging from the visible region to the near infra-red (IR) region due to its large absorption co-efficient [33–35]. In the visible spectrum region (above 1.57 eV), the absorption co-efficient of $CH_3NH_3PbI_3$ perovskite exceeds traditional inorganic absorption materials, such as GaAs, amorphous silicon (a-Si), and crystalline silicon (c-Si) (Figure 2b). By varying the cations and halide anions or the size of the nanocrystals, the absorption spectra can be tuned in the 400 to 800 nm region (Figure 2a). In lead (Pb)-based perovskite materials ($APbX_3$), the structure has a direct band gap between valence band and a conduction band. The conduction band (CB) is located at the 6p orbitals of Pb and the valence band (VB) is influenced by the halide X anions ($Cl^-$, $Br^-$, and $I^-$) component. Therefore, substituting $I^-$ to $Cl^-$ increases the band gap of perovskite, where absorption band gaps of 1.54 for $MAPbI_3$, 2.29 for $MAPbBr_3$, and 3.1 eV for $MAPbCl_3$ have been obtained. Likewise, since the ionic radius of Sn is smaller than that of Pb, substituting $Sn^{2+}$ to $Pb^{2+}$ increases the band gap. With the A cation, reducing the effective ionic radius results in an increase of the band gap (Figure 2c).

Based on the same principle, the emission spectrum of perovskite as a light emitter (e.g., LEDs and lasers) is tunable in wide range in the visible and near IR regions. By varying the halide anions, the emission wavelength of all inorganic perovskite $CsPbX_3$, where X is $Cl^-$, $Br^-$, $I^-$, and their mixture (Cl:Br or Br:I mixture), can cover all the visible range from 400 to 780 nm, which verifies the high possibility for R/G/B full color display and lighting applications [15,36,37].

Along with a tunable wavelength, all-inorganic perovskite $CsPbX_3$ (X = $Cl^-$, $Br^-$, and $I^-$) has exhibited high photoluminescence quantum yields (PL QYs about ~90%) and high color purity with narrow full-widths-at-half-maximum (FWHMs about ~20 nm) for all three primary colors, promising the realization of full color displays with a wide color gamut meeting the BT2020 standard [38–40] (Figure 2d). In addition, the single crystalline perovskite nanocrystals have shown narrower FWHMs for the emission peak compared to that of polycrystalline films due to reduced trap density states [41,42]. Benefitting from notable optical characteristics as well as their cost-effective solution processability, perovskite materials can be used as down conversion luminophores in backlit units of traditional displays such as liquid crystal displays or on the front glass of organic light-emitting diodes [43,44].

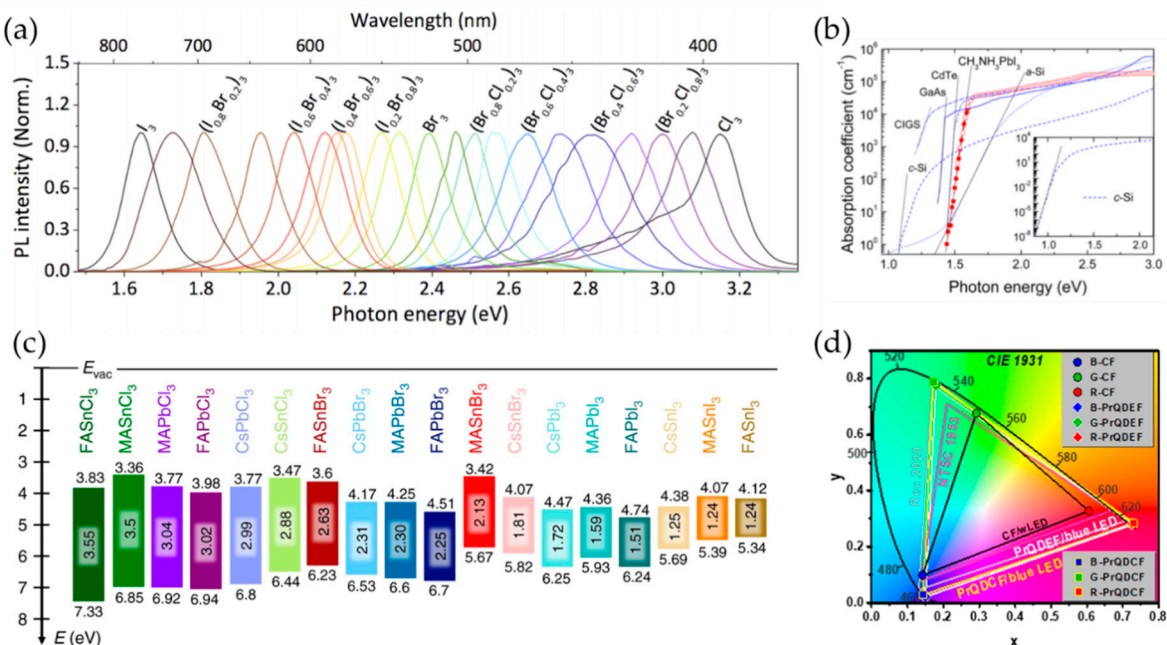

**Figure 2.** (**a**) Mixed halide (i.e., I, Br, and Cl) dependent tunable PL emission of $OA_{0.3}MA_{0.7}PbX_3$ (X = halide) perovskite crystals. Adapted with permission from [Perovskite Crystals for Tunable White Light Emission]. (**b**) Absorption coefficient of a $CH_3NH_3PbI_3$ perovskite crystals compared with other photovoltaic materials. Adapted with permission from [35] (**c**). Schematic diagrams of energy level of various perovskite crystals. Adapted with permission from [45]. (**d**) The CIE 1931 color space chromaticity diagram demonstrating the color coordinates of LCD using Perovskite QD color filter. Adapted with permission from [46].

### 2.2.2. Electrical Characteristics of Perovskite: Mobility, Lifetime, and Diffusion Length

The electrical characteristics (mobility, lifetime and diffusion length) of perovskite are key factors for determining the performance of optoelectronic applications (Table 1). The electrical properties of perovskite material have evolved enormously since the adaptation of the single crystalline structure. Since the single crystal perovskite has no grain boundaries, it exhibits largely decreased trap density states ranging from $10^9$ to $10^{11}$ cm$^{-3}$ [47–49], which is about several orders of magnitude lower than that of polycrystalline perovskite ($10^{15}$–$10^{19}$ cm$^{-3}$). These low trap density states produce ultralong charge carrier diffusion lengths, increased charge carrier mobility, and prolonged lifetime.

**Table 1.** Bandgap, carrier mobility, and diffusion length of halogen perovskite crystals.

| Perovskite Type | Bandgap (eV) | Carrier Mobility (cm$^2$/Vs) | Diffusion Length (μm) | Ref. |
|---|---|---|---|---|
| $MAPbI_3$ | 1.7 | 0.7–3 | 0.1–8 | [17] |
| $MAPbI_{3-x}Cl_x$ | 2.6 | 1.6 | 1–1.2 | [17] |
| $MAPbCl_3$ | 3.1 | ~40 | 3–9 | [50] |
| $MAPbBr_3$ | 2.3 | 20–100 | 3–17 | [49] |
| $FAPbI_3$ | 1.5 | 27 | 6.6 | [51] |
| $FaPbBr_3$ | 2.25 | 14 | 19.0 | [51] |

These trap density states are the main obstacles to achieving high luminescence efficiency (i.e., PL QYs), because they act as non-radiative recombination centers. The PL QYs of perovskite is defined by the ratio of the radiative recombination rate ($k_r$) to the overall recombination rate (PL QYs = $k_r/(k_r + k_{nr})$), where $k_{nr}$ is the non-radiative recombination rate [52]. Therefore, the suppressed trap density states of single crystalline perovskite show high PL QYs.

Similarly, the carrier lifetime is also highly affected by the non-radiative recombination lifetime, where the carrier lifetime ($\tau$) is expressed as following equation:

$$\frac{1}{\tau} = \frac{1}{\tau_r} + \frac{1}{\tau_{nr}} \tag{3}$$

where $\tau_r$ and $\tau_{nr}$ are the radiative and non-radiative recombination lifetime, respectively. Although the carrier lifetime is determined by the chemical structure, dimensions, and purity of the crystals, the decreased carrier lifetime indicates low crystallinity of the perovskite structure, which deteriorates the performance of optoelectronic devices such as solar cells and LEDs [53,54].

The carrier mobility and diffusion length are also critical factors for high performance perovskite-based energy harvesting applications (e.g., solar cells and photodetectors). The diffusion length ($L_D$) is defined as following equation:

$$L_D = \sqrt{\frac{K_B T \mu \tau}{q}} \tag{4}$$

where $K_B$ is the Boltzmann constant, T is the absolute temperature, and q is the elementary charge. Therefore, the single crystal MAPbX$_3$ (X = Cl$^-$, Br$^-$, and I$^-$) has a high diffusion length exceeding 1 μm, which is 1–2 orders of magnitude higher than that of polycrystalline perovskite material, and also high carrier mobility ranging from $10^2$ to $10^3$ cm$^2$/Vs [55,56].

### 2.3. Stability Issues in Perovskite Crystals

Although perovskite material shows outstanding performance in various optoelectrical parameters, the state-of-the-art perovskite suffers from poor chemical stability under ambient conditions [57–59]. It is well known that, when perovskite crystals are exposed to oxygen or moisture, the crystallinity of perovskite rapidly loses its phase and ultimately deteriorates the optoelectronic properties (Figure 3a). For example, the MAPbI$_3$ perovskite starts to decompose in the following steps when they are exposed to oxygen or moisture:

$$CH_3NH_3PbI_3 \leftrightarrow CH_3NH_3PbI \text{ (Aqueous)} + PbI_2 \tag{5}$$

$$CH_3NH_3PbI \text{ (Aqueous)} \leftrightarrow CH_3NH_2 \text{ (Aqueous)} + HI \tag{6}$$

$$4HI + O_2 \leftrightarrow 2I_2 + 2H_2O \tag{7}$$

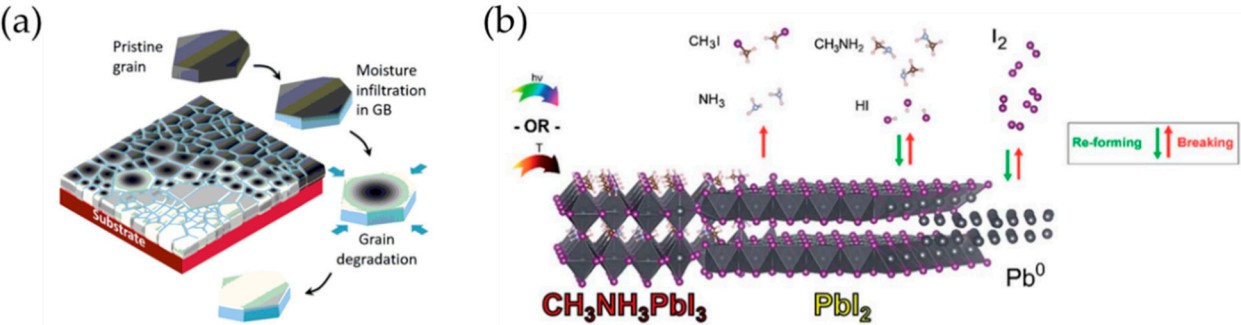

**Figure 3.** Schematic diagram of MAPbI$_3$ perovskite crystals degradation process (**a**) in presence of moisture. Adapted with permission from [60] and (**b**) in presence of photo- and thermal- induced conditions. Adapted with permission from [61].

This shows the decomposition process of MAPbI$_3$ perovskite to methylammonium lead iodide (MAI) and lead iodide (PbI$_2$). During the de-crystallization process, an intermediate complex such as [(CH$_3$NH$_3^+$)$_{n-1}$(CH$_3$NH$_2$)$_n$PbI$_3$][H$_3$O] can be formed which increases the defect states or the number of low optoelectronic property grains.

Therefore, single crystal perovskites, which have very low defect states and grain boundaries, can be one of suitable candidates for promising the chemical stability in moisture or air conditions. Hua-Shang Rao et al. fabricated single crystal MAPbBr$_3$ perovskite-based solar cells that exhibited high stability while maintaining a 93% initial PCE after aging for 1000 h in ambient conditions. Similarly, Zhipeng Lian et al. proposed single crystal MAPbI$_3$ perovskite-based photodetectors, which demonstrated excellent performance, $10^2$ times higher responsivity and $10^3$ times faster response speed compared to polycrystalline perovskite with moderate stability and durability.

Achieving light and thermal stability of perovskite are also challenging issues which can influence the performance of perovskite-based optoelectronic devices (Figure 3b). Photoinduced ion migration (e.g., MA$^+$ or Pb$^+$), which is accelerated by the presence of light, can damage the octahedron structure of perovskite or penetrate into adjacent layers, such as the hole-transport layer (HTL) and metal electrode. Thermal-induced phase transitions in perovskites are commonly observed. Since all the inorganic single crystal perovskites (e.g., CsPbBr$_3$) have shown for high thermostability up to 580 °C, the single crystal perovskite is attractive as a potential candidate for commercialization of perovskite applications in near future.

## 3. Perovskite Solar Cells

With continuously increasing population along with high criteria for future renewable energy targets, there is intensive needs for environmental-friendly energies such as solar cells. Exploiting the suitable properties of light absorbing materials, such as large absorption coefficients, long charge diffusion lengths, and high carrier mobility, perovskite-based solar cells (PSCs) have achieved noticeable progress within 10 years and reached a certified efficiency of 25.5% in a single junction and of 29.1% in perovskite/Si multijunction solar cells. Since the PSCs have shown outstanding performance and cost-effective solution processability, the solar cell technology is entering the fourth generation [62–66]. Industrial level research, such as large-scale production, environmental-friendly production, and stable operation, for the practical use of PSCs is also investigated widely in various research institutes (e.g., The Swiss Center for Electronics and Microtechnology (CSEM) and National Institute for Materials Science (NIMS)) [67–70], which have recently achieved efficiency of 26.5% in perovskite/Si multijunction solar cells in large-area production [71]. In recent, Oxford Photovoltaics (Oxford PV) achieved 29.52% of PCEs with the perovskite on GaAs solar cells which was also certified by the US National Renewable Energy Laboratory (NREL).

Since the first demonstration by T. Miyasaka et al. in 2009 [32,72], there have been intensive studies to improve the performance of PCSs. The structure of PSCs can be separated as a mesoporous n-i-p structure, planar n-i-p (conventional), and planar p-i-n structure (inverted) [72,73] (Figure 4a). These structures are meant to improve the charge extraction and transport characteristics, where the perovskite absorption layer is sandwiched between the electron-transport layer (ETL) and hole-transport layer (HTL). These charge transport layers (CTLs, both ETLs and HTLs) play crucial roles in extracting electron and hole charge carriers, which are generated from the perovskite absorption layer, and assisting the formation of a high crystallinity perovskite layer (Figure 4b). Therefore, along with the uniform distribution of surface properties, the energy level of CTLs needs to be located at a barrierless position rather than at perovskite absorption layers.

The most critical factor in determining the performance of the PSCs is the perovskite film property. Various studies have been reported regarding the morphological distribution of perovskite films for high uniformity, high crystallinity, and high phase purity of the perovskite structure. Therefore, the perovskite film fabrication process is the most basic and important factor which directly affects the film quality. This section discusses the role of CTLs and the perovskite film fabrication process in the performance of PSCs.

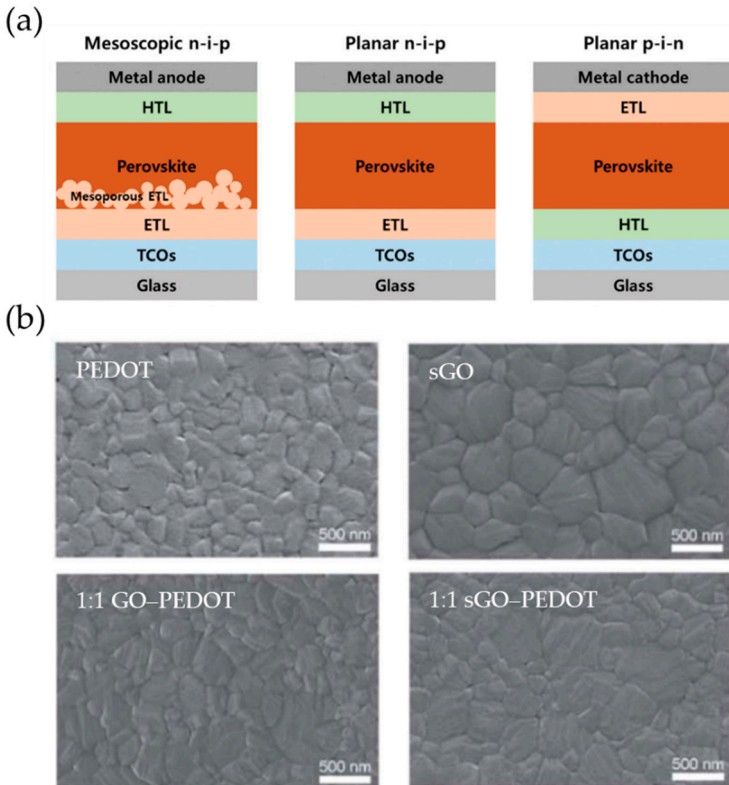

**Figure 4.** (**a**) Typical structures of the PSCs. Adapted with permission from [Atomic layer deposition for efficient and stable perovskite solar cells]. (**b**) SEM images of perovskite films deposited on the PEDOT, sGO, 1:1 GO–PEDOT, and 1:1 sGO–PEDOT film indicating effect of bottom charge transport layer on formation of perovskite crystals. Adapted with permission from [74].

### 3.1. Key Parameters for Solar Cells

The performances of solar cells are defined by the current density-voltage (J-V) characteristics under illumination. The main parameters, which indicate the performance of the device, are short-circuit current density ($J_{sc}$), open-circuit voltage ($V_{oc}$), fill factor (FF), and most importantly, the power conversion efficiency (PCE). The $J_{sc}$ is the current density when the devices are in zero bias conditions indicating the exciton dissociation property and charge transport property of the solar cells. Meanwhile, the $V_{oc}$ is the voltage difference measured between two electrodes when no current flows through the devices, indicating the built-in potential of solar cells. The FF and PCE are expressed as the following equations:

$$FF = \frac{P_{Max}}{J_{SC} \times V_{OC}} \tag{8}$$

$$PCE(\%) = \frac{\text{Output Power}(Electric)}{\text{Input Power}(Light)} = \frac{J_{sc} \times V_{oc} \times FF}{P_{light}} \times 100 \tag{9}$$

where $P_{Max}$ is the maximum power that the solar cells can produce and $P_{light}$ is the irradiated solar energy. The FF measures the "squareness" of the J-V characteristics in solar cells, which is directly affected by the series and shunt resistances of the devices. The PCE is measured in standard conditions, which is a temperature of 25 °C and a 1 sun irradiation (100 mW/cm$^2$) with an air mass of 1.5. The traditional silicon-based solar cells have shown an FF between 0.5 to 0.8 and a maximum PCE exceeding 25%.

### 3.2. Role of Charge Transport Layers

During the developmental progress of PSCs, much attention has focused on the development of the perovskite light harvesting layer. However, the CTLs are also crucial factors that determine both formation of perovskite film and performance of the devices. In the

early stage of developing PSCs, H. Kim et al. [75] adopted mesoscopic TiO₂ film as ETLs and 2,20,7,70-tetrakis-(N,N-di-4-methoxyphenylamino)-9,90-spirobifluorene (spiro-MeOTAD) as HTLs, achieving high efficiency (PCE ~ 10%) and stable PSCs (long-term stability tests for over 500 h). Their results emphasized the importance of CTLs in determining the performance of a PSC. There are several prerequisite factors [76,77] to function as ideal CTLs in PSCs: no energetic barrier for electrons and holes, high thermal stability, and non-toxicity. Until now, various ETLs (e.g., TiO₂, SnO₂, and ZnO) and HTLs (e.g., Spiro-OmeTAD, NiO, CuO, and PTAA) have been reported to achieve high performance PSCs.

The most important characteristic of CTLs is the energy level alignment with the perovskite absorption layer (Figure 5a). For the energetically barrierless device configuration, the lowest unoccupied molecular orbital (LUMO) of the ETL should be located at a lower energy state than the perovskite layer and the highest occupied molecular orbital (HOMO) of the HTL should be located at a higher energy state than the perovskite layer. Moreover, the transmittance of CTLs in the UV-Vis region should be taken into consideration to maintain the number of photons during light exposure. The bottom layer of perovskite should have a smooth and uniform surface to obtain high crystallinity of the perovskite film, and the upper layer of the perovskite should use non-polar solvent not to damage the perovskite layer.

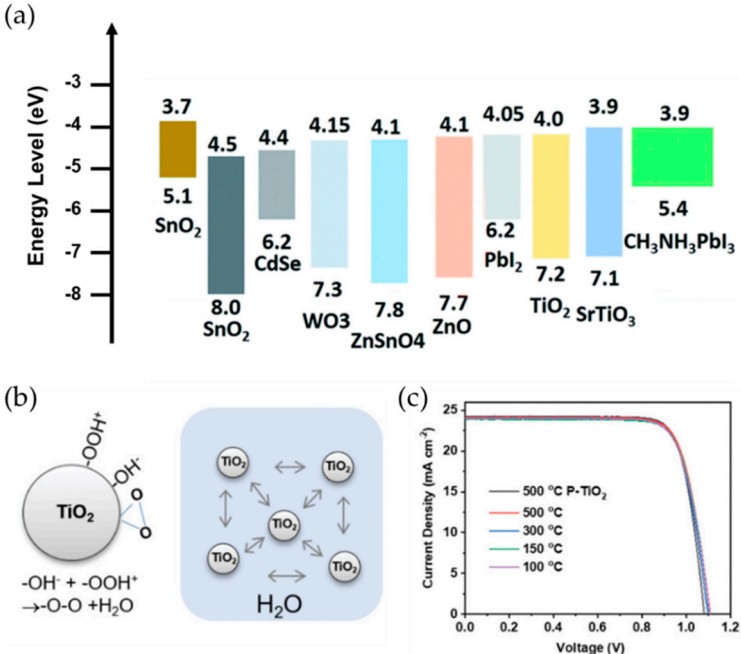

**Figure 5.** (**a**) Energy band diagram of various ETLs with respect to MAPbI₃ perovskite crystals. Adapted with permission from [76]. (**b**) The preparation of a TiO₂ ETL for high performance PSCs at low temperatures based on the synthesis of a stable TiO₂ colloidal aqueous solution. (**c**) Current density−voltage characteristics of PSCs fabricated by TiO₂ ETLs with varying annealing temperature. Adapted with permission from [78].

By taking all these considerations, transition metal oxide TiO₂ is the most widely used for high performance PSCs [78]. Since TiO₂ shows good chemical stability, a suitable energy level position compared with perovskites, and cost-efficient solution processability, various studies have reported the successful implementation of TiO₂ as the ETL in PSCs. Recently, M. J. Paik et al. [79] proposed the high quality TiO₂ layer by combining the TiO₂ underlayer, which was formed by chemical bath deposition, and sprayed TiO₂ colloids (Figure 5b). The dense and uniform TiO₂ ETL-based PSCs have shown high performance with a PCE of 22.7%. Various studies have also widely investigated TiO₂ with a bilayer ETL (e.g., TiO₂/WO₃, TiO₂/SnO₂, and TiO₂/ZnO) [79–81] or applying metal doped TiO₂ ETL (e.g., Li, Al, Nb, and Ni) to improve the optoelectrical property of the TiO₂ ETL. Although a

high annealing temperature has been raised as a challenging issue in the $TiO_2$ ETL, various low temperature processable $TiO_2$ ETLs with alternative fabrication methods (e.g., atomic layer deposition (ALD) and E-beam) have been reported [82–84].

### 3.3. Deposition Method of Perovskite Film

Two deposition processes have been most widely investigated in perovskite film fabrication: Solution process and thermal evaporation deposition process (Figure 6a,b). In this section, we discuss perovskite film fabrication methods (solution process and evaporation process) and their effect on morphology and performance in PSCs.

#### 3.3.1. Solution Process

In the solution spin coating process, one step and two step spin coating processes are the most commonly used in solution processes of perovskite film fabrication (Figure 6a). The one step spin coating process has the advantages of simplicity and low-cost processability, which is conducted using a perovskite precursor solution which is dissolved in polar solvents (e.g., DMF, DMAc, DMSO, and GBL) [85,86]. During the spin coating process and annealing process, the crystal structure of perovskite is determined by ionic interaction between anions and cations in the precursor. However, due to the slow crystallization process, partial aggregations and pinholes are generated, which results in low reproductivity in the film formation.

To enhance the uniform distribution of the perovskite film quality, a two-step spin coating process was proposed by K. Liang et al. [87] who used a two-step solution deposition method. They successfully fabricated $CH_3NH_3BI_3$ (B = Pb, Sn) perovskite films by depositing $BI_2$ film in advance and converting the film into a perovskite layer by adopting a $CH_3NH_3I$ (MAI) solution. These sequential steps of $BX_2$ and AX were investigated. J. Im et al. [88] reported high performance $CH_3NH_3PbI_3$ PSCs with varying the MAI concentration. The concentration of MAI highly influences the cuboid size of perovskite films, which influences the optoelectronic characteristics of the perovskite absorption layer by altering the packing density of the crystal population and the light harvesting property. Therefore, by optimizing the cuboid size of $MAPbI_3$ perovskite films, the PSCs achieved a high level of light harvesting and charge carrier extraction properties resulting a PCE of 17%. Similar studies have been reported on enhancing the enhancing film morphology. H. Ko et al. [89] and N. Ahn et al. [90] reported the effect of $PbI_2$ concentration and annealing temperature in grain growth of perovskite film, resulting in a PCE of 15.8% and 14.3%, respectively (Figure 6c,d). Later, the crystal growth mechanism of perovskite in two step fabrication was quantitively investigated by Y. Fu et al. [91,92]. By adopting sufficient concentration of MAI in $PbI_2$ films, the interface reaction initially formed a $MAPbI_3$ structure along with producing $PbI_4^{2-}$. After the saturation amount in the MAI solution, the $PbI_4^{2-}$ reacted with $CH_3NH_3^+$ and slowly recrystallized to form a single crystal $MAPbI_3$ perovskite structure. However, along with incomplete conversion of $PbI_2$ residues, a long reaction time still remained as challenging issues which easily resulted in uneven formation of large crystals.

In both one step and two step processes, controlling the nucleation and crystal growth is highly emphasized for uniform formation of perovskite film morphology. One efficient way to solve this issue is inducing an anti-solvent during the formation of perovskite crystals. M. Xiao et al. [93] deposited chlorobenzene (CBZ) as an anti-solvent solution during one step spin coating of $CH_3NH_3PbI_3$ (dissolved in DMF solvent) and achieved large crystalline perovskite grains with micron size. Compared to the conventional one step process, the PSCs with that used the anti-solvent process showed 192% enhancement in PCE. The role of the anti-solvent during the spin coating process controls the crystallization process by removing the residual precursor and enhancing uniformity of the perovskite film. Various kinds of solvents are adopted in anti-solvent processes. D. Son et al. [94] and N. J. Jeon et al. [95] used diethyl ether on $MAPbI_3$ and toluene on $FAPbI_3$, respectively, which showed remarkable improvement in performance of PSCs. With a similar princi-

ple, a mixture of anti-solvent was also investigated to form high quality perovskite film. J. Liu et al. [96] used a mixture of ethyl acetate (EA) and isopropyl alcohol (IPA) for spin coating a MAPbI$_3$ perovskite precursor and achieved high PCE of 18.9%.

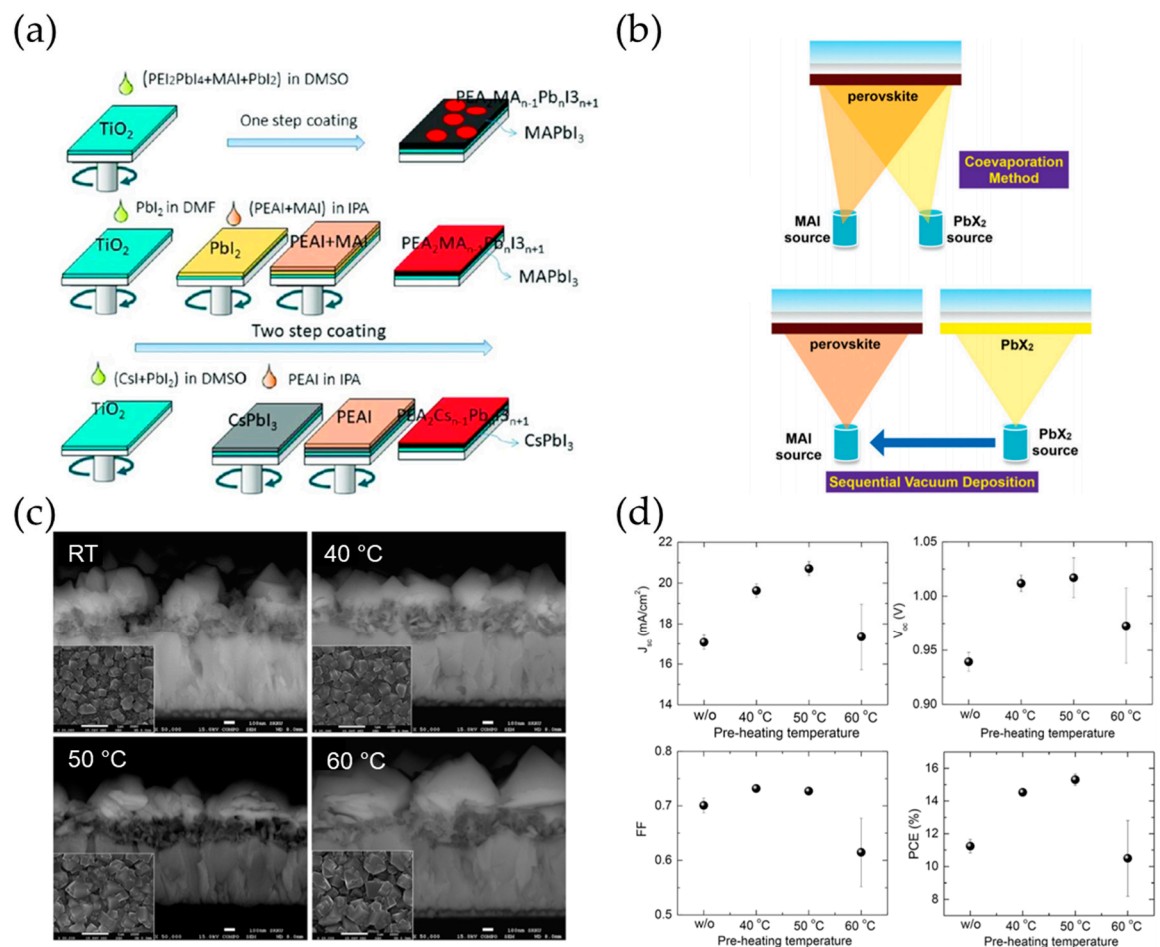

**Figure 6.** (**a**) Typical solution process (One step and two step) of perovskite crystals. Adapted with permission from [Stability Issue of Perovskite Solar Cells under Real-World Operating Conditions]. (**b**) Schematic illustration of perovskite fabrication by using thermal evaporation process. Adapted with permission from [97]. Effect of pre-heating temperature on (**c**) the grain size of perovskite and (**d**) the PSCs performance. Adapted with permission from [89].

### 3.3.2. Thermal Evaporation Process

Fabricating a perovskite layer by the solution process inevitably accompanies a thermal annealing process on the substrates. However, the thermal annealing process limits the feasibility of choosing a substrate, which is especially essential in flexible optoelectronic devices. The thermal evaporation process can solve this issue and is also an efficient method to fabricate uniform and smooth perovskite film. Similar to the solution process, two methods are widely used to fabricate perovskite film via thermal evaporation: single source and dual source (or multiple-source) evaporation.

In the thermal evaporation process, the dual source evaporation method was reported earlier by M. Liu et al. [98] They presented CH$_3$NH$_3$PbI$_{3-x}$Cl$_x$ mixed halide-based PSCs by evaporating MAI and a PbCl$_2$ precursor in dual source simultaneously. The obtained perovskite film had a smooth and uniform morphology and the PSCs achieved a high PCE exceeding 15%. C. Meanwhile, Chen et al. [99] and S. Hsiao et al. [100] proposed CH$_3$NH$_3$PbI$_3$-based PSCs in sequential steps of PbI$_2$ and MAI in the thermal evaporation process, achieving perovskite film with a fine morphology. Since all inorganic perovskite (e.g., CsPbI$_3$ and CsPbI$_x$Br$_{1-x}$) [101–103] has also been demonstrated by utilizing thermal

evaporation, the thermal evaporation process attracted strong attention as an alternative method of solution processing.

## 4. Other Applications

### *4.1. Photodetectors*

Photodetectors are also strong candidates for future applications, including thin film solar cells and light emitting diodes. Organic-inorganic hybrid perovskite structures are suitable for photo-sensing applications with their extraordinary characteristics, such as low binding energy, high mobility, high dielectric constant, and ambipolar charge transport. Compared to the conventional types of photodetectors, such as Si and group III-V semiconductors, the perovskite structures exhibit large absorption coefficients ($\sim 10^5$ cm$^{-1}$) [104] over a wide range of wavelengths. Thanks to these characteristics, the thin layer is sufficient for the high absorption of the perovskite photodetectors. The thin active layer is favorable for achieving high electrical characteristics in the devices because the drift distance of the photogenerated carrier is small. The short travel length of the photocarrier decreases the recombination rate and increases the response speed. With these many advantages of the organic-inorganic hybrid perovskite structures, the photodetector applications have deserved the attention of the many researchers involved in the industries concerned with environmental monitoring and biomedical sensors.

For high performance photodetectors, the perovskite photodetectors need to exhibit detection of weak light under small noise signals as well as high photocurrent under light illumination. In the organic photodetectors and organic-inorganic hybrid photodetectors, the dark current is dominant for noise so the reduction of the dark current is one of the key issues in perovskite photodetectors. The reduction of the dark current and the enhancement of the driving current under illumination create a high degree of detection in the device. Such key parameters for the perovskite photodetectors are discussed in the following subchapter. Recently, the perovskite photodetectors emerged as a self-powered photodetector which operates the device without any external bias. With a long mean free path of the charge and thin active layer, the photogenerated charge efficiently collected to the both electrodes without external bias [105,106]. The self-powered applications have many advantages: simple device structures, reduced fabrication steps, and triboelectric nanogenerators (TENGs) applications [107–109]. According to the many advantages, TENGs based on perovskite photodetectors have use ranges in remote areas, including outdoor environments and mobile applications.

### 4.1.1. Key Parameters for Photodetectors

The performance of the photodetectors can be evaluated by several figure-of-merit parameters: specific detectivity, responsivity, and linear dynamic range. The parameters digitize the features of the device, namely sensitivity, efficiency, and measurement range. The specific detectivity (D*) is the parameter for the ability of the device how weak a light the device can detect. The specific detectivity is calculated by Equation (10).

$$D^* = \frac{(AB)^{\frac{1}{2}}}{\frac{i_n}{R}} \qquad (10)$$

where A is the device area, B is the electrical bandwidth, and in is the noise current integrated through the whole system. Generally, the noise occurs by four prime factors: shot noise ($i_s$), thermal noise ($i_t$), 1/f noise ($i_{1/f}$), and generation-recombination noise ($i_{g-r}$). The noise current can be expressed as the combination of these four factors.

$$i_n = \sqrt{i_s^2 + i_t^2 + i_{1/f}^2 + i_{g-r}^2} = \sqrt{2eI_dB + \frac{4KTB}{R_{sh}} + i(f,B)_{1/f}^2 + i(f,B)_{g-r}^2} \qquad (11)$$

where B is the bandwidth, K is the Boltzmann constant, T is the temperature, and $R_{sh}$ is the shunt resistance. However, the noise current of the photodetecting system including the perovskite structure is determined by the dark current of the device because the dark current is much larger than the other factors for noise. Therefore, D* is given by Equation (12).

$$D^* = \frac{J_{light}}{P_i(2qJ_d)^{\frac{1}{2}}} \tag{12}$$

where $P_i$ is the intensity of the incident light, q is the quantity of the charge, $J_d$ is the dark current, and Jlight is the current under light illumination. While the detectivity represents the sensitivity of the photodetector devices, responsivity represents the conversion efficiency of the device. The responsivity (R) of the photodetector is expressed by Equation (13).

$$R = \frac{J_{ph}}{P_i} = \frac{J_{light} - J_d}{P_i} \tag{13}$$

where the parameters in Equation (4) are identical to the parameters in the equation of D*. As R represents the conversion efficiency from the light to the electrical current, R has an intimate relation to the external quantum efficiency (EQE). R can also be expressed by the EQE divided by the photon energy of the light.

### 4.1.2. Progess in Perovskite Photodetectors

Since Dou et al. first reported solution-processed hybrid perovskite photodetectors in 2014 [110], many researchers have developed high performance photodetectors using perovskite structures. The improvements of the photodetectors focused on increasing the working current, reducing the dark current, and increasing the response speed. Considering the conventional photodetector devices such as GaN, Si, and InGaAs, the perovskite photodetectors showed comparable specific detectivity and linear dynamic range (LDR). The device consisted of thin film structures with glass/ITO/PEDOT:PSS/$CH_3NH_3PbI_{3-x}Cl_x$/PCBM/PFN/Al, in which PFN served as the extraordinary hole blocking for the devices. With the excellent characteristics for blocking the charge transport when the device was under no light, the dark current was decreased to $1.5 \times 10^{-11}$ mAcm$^{-2}$. The fabricated device showed a high detectivity of $10^{14}$ Jones. Likewise, reduction of dark current is one of the most important factors to achieve high detectivity (see Equations (9)–(11)). The effective strategy to reduce the dark current is inception of charge blocking layers. Fang et al. [111] introduced the OTPD as the electron blocking material and $C_{60}$ as the hole blocking material (Figure 7a–c). Generally, PEDOT:PSS is widely used as hole injecting materials for the perovskite photodetectors whose electron injection barrier between the ITO is 1.1 eV. The OTPD increased the electron injection barrier to 2.5 eV at the interface, which suppressed the electron transport under dark conditions. $C_{60}$ was deposited on the opposite side of the device by thermal evaporation. Because of the conformal deposition of thermal evaporation, the $C_{60}$ layer filled the pin holes of the perovskite layers with rough morphology. The $C_{60}$ layer also had a large hole injection barrier of 2.0 eV that suppressed the hole transport within the electron transporting site. To introduce the large single carrier conducting characteristics at the both sides, the perovskite photodetector was fabricated with the structure glass/ITO/OTPD/perovskite/PCBM/C60/BCP/Al. The noise current of the device was reduced to $10^{-15}$ AHz$^{-0.5}$, which was a level similar to the shot noise limit or the thermal noise limit. CuPc [112], MoS2 [113], PBDB-T:IHIC mixed layer [114], Zr-doped TiOx [115], P3HT [116], and so on are used to reduce the dark current of the perovskite photodetectors.

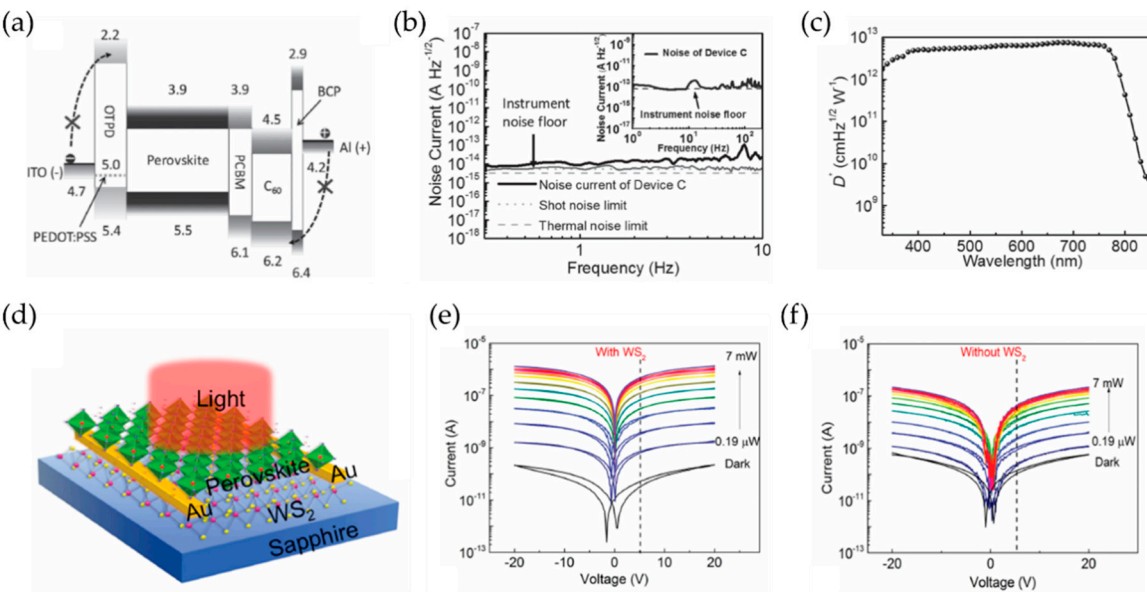

**Figure 7.** (**a**) The energy diagram of the perovskite photodetectors with the double fullerene layer, (**b**) The measured total noise current from 0.3 to 10 Hz under −0.1 V bias (black solid line), the calculated shot noise limit (gray dotted line) and thermal noise limit (gray dashed line), and the instrument noise floor (gray solid line). The inset is the measured total noise current with a wider frequency range (solid line) and the instrument noise floor with such bandwidth (dashed line) which clearly shows that the noise at higher frequency is lower than the equipment detection limit. (**c**) The specific detectivity at different wave-length of light under −0.1 V bias. Adapted with permission from [111] (**d**) Schematic device structure of the hybrid $WS_2$/perovskite photoconductor fabricated on a C-plane (0001) sapphire substrate, (**e**) I–V curves of the hybrid $WS_2$/perovskite bilayer photoconductor measured in dark and under illumination with different white-light intensities, and (**f**) corresponding I–V curves measured on the perovskite single layer. Adapted with permission from ref [117].

To reduce the dark current, it was also important to increase the crystallinity of the perovskite structure as well as considering the device design for the charge blocking. Ma et al. reported the suppressed dark current of the perovskite photodetectors toward enhanced detectivity [117]. They synthesized a $WS_2$/$CH_3NH_3PbI_3$ heterostructure as the active layer of the photodetectors. The polycrystalline $WS_2$ monolayer was formed by a chemical vapor deposition (CVD) method and the perovskite layers were grown on the $WS_2$ layer. The perovskite layers grown on the $WS_2$ monolayer increased the crystallinity, which was observed by photoluminescence and Raman analysis. The improvement of the crystallinity created not only a high photocurrent, but also a low dark current because one of the major causes of dark current existed along the grain boundary of the perovskite structure. The fabricated perovskite photodetector exhibited a high on/off ratio ($\sim 10^{15}$), high responsivity ($\sim 17\ AW^{-1}$), and a fast response speed (a time constant of 2.7 ms) (Figure 7d–f). Saidaminov et al. [118] reported a planar-integrated single-crystalline perovskite structure to achieve high detectivity in the photodetectors. The grain size of the crystalline perovskite structure was increased over 30 μm using a solvent engineering process. The solvent engineering controlled the speed of crystal formation that resulted in characteristics of crystallinity. The integrated single-crystalline perovskite structure showed almost identical photophysical characteristics of free-standing crystals for the charge mobility, exciton lifetime, and diffusion length. As a result, the integrated single-crystalline perovskite photodetectors showed a high responsivity of $\sim 4000\ AW^{-1}$ in a wide wavelength range from UV to visible green light.

Though the device structures of photodetectors are similar to photovoltaic cells, photodetectors need to satisfy additional requirements, such as a fast response time. In this common interest, many researchers have focused on the response time as well as the detectivity or responsivity. Li et al. [106] reported self-powered photodetectors with a high response speed using a CsPbBr3:ZnO active layer. Instead of $CH_3NH_3^+$ organic cation,

Cs is used for all-inorganic perovskites. The all-inorganic perovskite structures have the advantages of high thermal stability, high electron mobility and small exciton binding energy. Furthermore, ZnO nanoparticles (NPs) were mixed into the perovskite structures where the photogenerated electron-hole pairs were successfully separated. Compared to the $CsPbBr_3$ perovskite layers, the $CsPbBr_3$:ZnO layer showed decreased characteristics of absorption and photoluminescence. However, the $CsPbBr_3$:ZnO photodetectors had better charge extraction characteristics under no external bias, which resulted in a fast rise time of 18 ms. Shen et al. [119] also reported a high-speed response for the all-inorganic perovskite solar cells with the structure glass/ITO /PCBM/PTAA/PEIE/$CsPbI_xBr_{3−x}$/BCP/Ag. PEIE layers modified the PTAA surface, which improved the film quality. The reduced trap sites of the all-inorganic perovskite structures increased the response time and improved the device stability [120]. The fabricated photodetectors with all-inorganic perovskite structures showed a 20 ns response time and 2000 h lifetime. Carbon nanotubes (CNT) [121], interdigitated (IDT) [122] patterned Au, $PdSe_2$ [123] was used to achieve fast response of perovskite photodetectors.

### 4.2. Light Emitting Diodes

Perovskite light emitting diodes (PeLED) are devices used for conversion of electrical energy to light, whereas photovoltaics and photodetectors are for the conversion of light to electrical energy. Tan et al. [3] first reported the application of perovskite structures to the light emitting diode application in 2014. They introduced the electroluminescence of a $CH_3NH_3PbI_{3−x}Cl_x$ perovskite structure with a near infrared wavelength (754 nm) and narrow emission band (35 nm of full-width at half-maximum). Since then, many researchers have studied the outstanding photoluminescence or electroluminescence characteristics of perovskite structures. The perovskite structures have adequate characteristics for the light emitting devices in high photoluminescence quantum yield (PLQY), low temperature processability, narrow bandwidth, and high electrical mobility. The emission peak of the perovskite structures can be easily tuned by simply substituting a halide anion (X of $ABX_3$ structure) to another halide anion, such as $I^-$, $Br^-$, and $Cl^-$. Kumawat et al. [124] reported the bandgap tuning of $CH_3NH_3Pb(Br_{1−x}Cl_x)_3$ perovskite structure by varying x from 0 to 1. The bandgap of the perovskite film was tuned from 2.42 eV to 3.16 eV that covered ultraviolet (409 nm) to visible light. Eperon et al. [125] controlled the bandgap of $FAPbI_yBr_{3−y}$ perovskite structure by varying y from 0 to 1 which covered from visible to near infrared (NIR) (840nm). These wavelength tunability of perovskite structure under compositional control enables the color controllability in the full range of visible light, which creates the key advantages in full color or white displays, lighting applications, and so on. Also, wide tunable perovskite structures can expand their applications to UV LEDs and NIR LEDs.

There are versatile strategies to realize perovskite-based displays in approaches based on morphology control. Many morphological types of perovskite structures, such as three-dimensional structures [126], quantum dots [127], nanowires [128], two-dimensional layers [129] and nanocrystals [37,129] were developed and introduced for optoelectronic devices. The dimensionally confined nanostructures of the perovskite materials exhibit a uniform morphology as well as a large surface area and anisotropic characteristics. The low-dimensional perovskite structures were utilized in many applications, such as solar cells, photodetectors, and phototransistors. Among them, light emitting diodes were the most studied application because of their unique advantages of high color purity and compatible tunability.

### 4.2.1. Key Parameters for Light Emitting Diodes

The crucial factors in evaluating light emitting diodes are concerned with the conversion efficiency from electrical energy to light. Three parameters are mainly used for device

efficiency: external quantum efficiency (EQE), current efficiency (CE), and power efficiency (PE). These parameters are calculated using following Equations (14)–(16).

$$EQE = IQE \times \eta \tag{14}$$

$$CE = L/J \tag{15}$$

$$E = P/IV \tag{16}$$

where IQE is the internal quantum efficiency, η is the light extraction factors, L is the luminance, J is the current density, P is the luminous flux, I is the current, and V is the voltage. EQE represents the ratio of the number of emitted photons and the number of injected electrons. CE represents the ratio of the luminance of the light and the injected charge. PE represents the ratio of the emitted light intensity and the power consumption.

Not only the efficiencies of the light emitting diodes, but the lifetime (LT80, LT50), maximum luminance ($L_{max}$), and turn-on voltage ($V_{on}$) are also crucial parameters for the devices. LT80 (LT50) is the time when the luminance is decreased to the 80% (50%) of the initial luminance. The turn-on voltage is the voltage where the luminance reaches 10 Cd/m$^2$.

### 4.2.2. Three-Dimensional Perovskite Structures for Light Emitting Diodes

Light emitting diodes with three-dimensional perovskite emissive layers were developed in both normal structures and inverted structures. The anode is the bottom electrode in a normal structure and the cathode is the bottom electrode in an inverted structure. When the device structures are designed, PEDOT:PSS [130], TFB [131], poly-TPD [132], and so on are selected as hole transporting layers and ZnO [133], TiO$_2$ [134], TPBi [135], and so on are selected as electron transporting layers. The most critical issues in the perovskite light emitting diodes are the leakage current resulting from the nonuniform morphology of the perovskite structures and the trap sites at the interfaces between the layers. Lin et al. [13] reported perovskite light emitting diodes with a high EQE (~20%) using leakage reduction and trap passivation with a MABr layer. They designed the device structures as ITO/PEDOT:PSS/CsPbBr$_3$/MABr/B3PYMPM/LiF/ Al. The MABr layer was deposited between the emitting layer and the electron transporting layer by mixing additional MABr in the precursor solution. With the mixed precursor solution, the perovskite films were formed in a vertically graded composition because of the difference in solubility. The efficiency of the light emitting diodes were increased more by adopting PMMA for electron-hole charge balance. Kang et al. [136] also reported the reduction of leakage current with a bilayer structure of the perovskite emitting layer. Before the CsPbBr$_3$ emitting layer was deposited, the MAPbCl$_3$ layer was deposited as a hole transporting layer. The MAPbI$_3$ layer exhibited low density in the trap sites, chemical resistance from the CsPbBr$_3$ layer deposition, and increased hole extraction characteristics that improved the EQE at 5.04% (Figure 8a,b). Furthermore, many small organic molecules such as polyethylene oxide (PEO) [137], 2-((4-biphenylyl)-5-phenyl-1,3,4-oxadiazole) (PBD) [138], [2,2-(ethylenedioxy)bis(ethylammonium), EDBE], and so on was studied for leakage engineering of perovskite light emitting diodes.

Another strategy to increase the device efficiency is grain size control to increase the coverage and improve the photophysical characteristics. Cao et al. [139] reported the size control of a perovskite structure with amino acid additives. The grain size was controlled in a submicrometer scale which controls wave-guiding of the emitted light and defect passivation on the perovskite surface. The interface between the electron transporting layer and the perovskite layer was also engineered by polyethylenimine ethoxylated (PEIE) for further leakage reduction and electron-hole balance. Solvent engineering is also a good approach to increase uniformity and coverage. Zhang et al. [140] reported the chlorobenzene-assisted pre-annealing process (CPA) treatment for the perovskite structure formulation. The solvent engineering decreased the evaporation rate and increased the drying time of the perovskite layer. As a result, the compact perovskite film was formed

with reduced porosity, which produced a large effective area and a high EQE of 11.05%. Not only is the increase of photophysical characteristics of the emitting layer important, but the charge injection properties of the hole transporting layer and electron transporting layer are also important to achieve high efficiency. In most cases, charge carrier mobility of the anode side is lower than that of the cathode side because metal oxide semiconductors such as ZnO and $TiO_2$ are generally used as the electron transporting layer. Therefore, the increase in hole injection from the anode side contributes the electron-hole balance as well. Zhao et al. [141] reported the comparison of TFB and poly-TPD as a hole transporting layer of a perovskite light emitting diode. Even though the highest occupied molecular orbital (HOMO) level of TFB was more matched with the HOMO of the $FAPbI_3$ perovskite layer than that of poly-TPD, the device performance was higher at the device with a poly-TPD hole transporting layer because of the shallow ionization potential of poly-TPD. Zhao et al. [142] also reported the outstanding hole injection properties of poly-TPD in perovskite light emitting diodes, and they additionally improved the layer through p-doping with F4-TCNQ. On the other hand, the electron transporting layer of $POPy_2$ was doped with $[RuCp*Mes]_2$ and the fabricated device showed a high EQE of 17%. They also revealed that Joule heating of the device is one of the most critical issues to achieve highly efficient and long lifetime light emitting diodes. The performance of the device was compared according to the different substrates. All of the parameters of the device, namely, the radiance, efficiency, and lifetime under high current density, increased at the sapphire with graphite where it is easy for the heat to be dissipated to the outside.

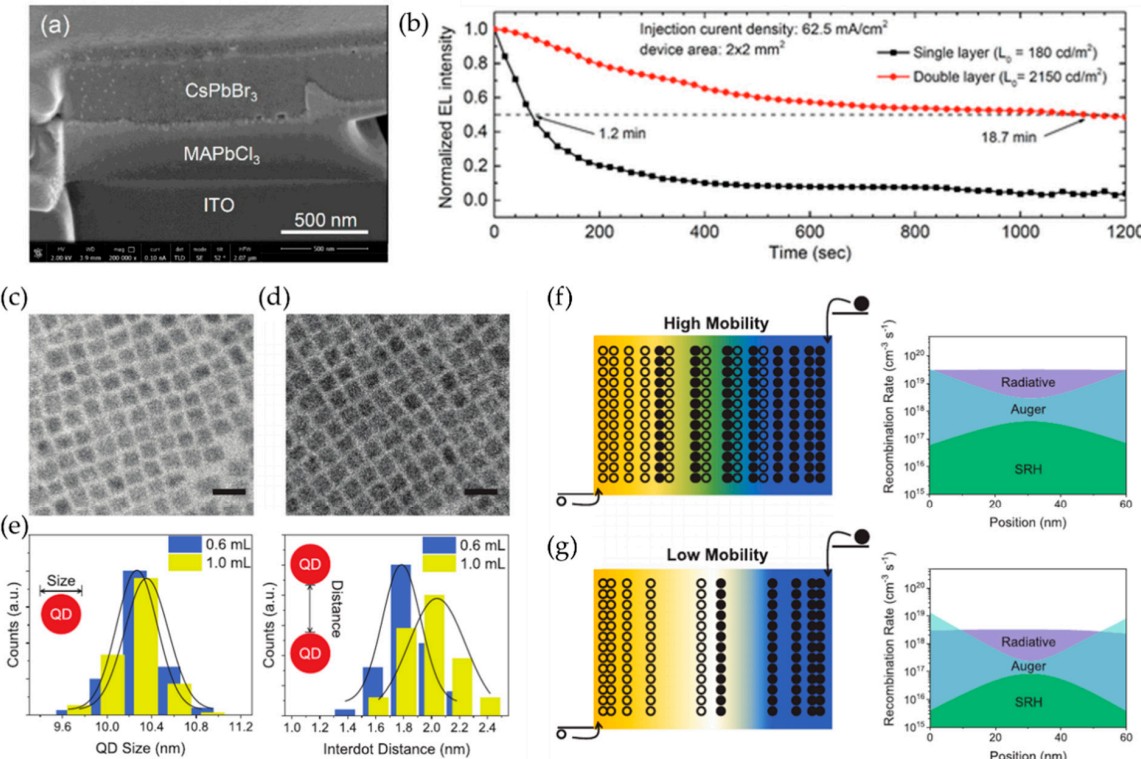

**Figure 8.** (**a**) FIB-assisted cross-sectional SEM image of the double-layer structure. (**b**) Normalized electroluminescence (EL) intensity versus time, measured under an injection current density of 62.5 mA/cm². The device area was $2 \times 2$ mm². Initial luminance ($L^0$) was 180 cd/m² for the single layer and 2150 cd/m² for the double layer. Adapted with permission from [136] (**c**), (**d**) TEM images of QDs synthesized with 0.6 and 1.0 mL oleyamine. Scale bars: 20 nm. (**e**) Distributions of QD size and interdot distances of QDs with different OLA amount. Schematic diagrams for carrier distributions in active layer and recombination rates of active layers of 0.6 mL and 1.0 mL of oleyamine, respectively. Schematic diagrams of carrier distributions and recombination rates of active layers of (**f**) 0.6 mL and (**g**) 1.0 mL of oleyamine. Adapted with permission from [143].

### 4.2.3. Low-Dimensional Perovskite Structures for Light Emitting Diodes

Two-dimensional perovskite structures have a modified chemical composition from the three-dimensional perovskite as the basic formula $A'_2A_{n-1}MX_{3n+1}$. In this formulation, A' is an organic radical of a long chain that induces separation of perovskite structures into planar structures. Mitzi et al. [144] first reported the two-dimensional <110>-oriented perovskite sheets with the chemical structure of $(C_4H_9NH_2)_2(CH_3NH_3)_{n-1}Sn_nI_{3n+1}$ (n = 1~5). The perovskite sheets were obtained by adding $SnI_2$ to the precursor solution. The formulated perovskite sheets exhibited different electrical characteristics, and the semiconducting property of the monolayered perovskite was changed to metallic property by increasing the number of the layers. This precise band gap tunability has advantages in display applications because the band gap tunability creates high color purity. Tian et al. [145] reported red light emitting diodes with quasi-2D perovskite structures. The composite of quasi-2D perovskite and poly(ethylene oxide) (PEO) is used as the emitting layer. The PEO played the role of layer planarization and trap passivation. The emission of the perovskite structure varied from 638 nm to 690 nm according to the number of the layers. With a structure of ITO/PEDOT:PSS/poly-TPD/PEO:perovskite/BCP/LiF/Al, the device exhibited an EQE of 6.23%.

The first zero-dimensional perovskite nanostructures were reported by Kojima et al. [146] in 2012. A zero-dimensional morphology was obtained on the mesoporous $Al_2O_3$ templates. Along with the nano-sized porosity of the $Al_2O_3$ film, the perovskite nanostructure was crystallized in the shape of nanoparticles. Since then, many methods to synthesize perovskite quantum dots (QDs) or nanocrystals (NCs) have been developed, such as hot injection methods, nanocrystal spinning, as well as the templated methods. One of the most important engineering approaches for highly efficient light emitting diodes is the quantum well designing of perovskite QDs emitting layers. Pan et al. [147] reported a ligand exchange of perovskite QDs from oleylamine and oleic acid as commonly used capping agents to di-dodecyl dimethyl ammonium bromide (DDAB), a relatively short ligand. The perovskite quantum dot films with a short ligand exhibited easy charge transport in the QD films. They introduced the PeLED with the structure ITO/PEDOT:PSS/PVK/perovskite QDs with short ligand/TPBi/LiF/Al and achieved a high current efficiency of 42.9 Cd/A and EQE of 8.53%. Quantum well engineering could also be performed by control of interdot distance. The interdot distance controlling was studied by the concentration of oleic acid in the precursor solution [143]. While the interdot distance was 1.99 nm from the perovskite quantum dot with 1ml of oleic acid, the distance was decreased to 1.82 nm with 0.6 mL of oleic acid. The close distance between the QDs suppressed luminescence quenching caused by Auger recombination, which resulted in a high EQE of 12.7%. Likewise, the photophysical characteristics of the perovskite QDs emitting layers are affected by many variables such as crystallinity, the length of the ligand, the amount of the ligand, and so on (Figure 8c–g).

As three-dimensional perovskite light emitting diodes carefully engineered trap sites between the layers, trap passivation of the perovskite QD layers was also researched to achieve high efficiency. Lu et al. [148] reported the zirconium-mediated surface modification of perovskite QDs. Zirconium acetylacetonate was introduced during the synthesis of perovskite QDs. The Zr-modified perovskite showed reduced net recombination because of the surface trap passivation. The device structure was designed as Ag/ZnO/PEI/perovskite QDs/TcTa/MoO$_3$/Au with the top emitting microcavity effect that exhibited a high EQE of 13.7%. Xu et al. [149] reported trap passivation on the double side of the QD layer. Diphenylphosphine oxide-4-(triphenylsilyl)phenyl (TSP01) was coated before and after the coating of the QDs. With the bilateral trap passivation, the PLQY of the QDs increased from 43 to 79% and the EQE increased from 7.7 to 18.7%. The lifetime of the device was also 20 times longer than the device without the passivation. Quaternary ammonium salts [3], organic–inorganic hybrid ligand [150], trioctylphosphine oxide (TOPO) [151], ammonium thiocyanate [152] and so on were introduced as a trap passivation layer for the perovskite QDs light emitting diodes.

Despite these extraordinary characteristics of the perovskite light emitting diodes, the commercialization of these device remains challenging due to the insufficient stability. Not only the light emitting devices, all the optoelectronic devices with perovskite structures are facing stability issue but low-dimensional perovskite structures for light emitting diodes are having more difficulty on it. The display needs a long lifetime over $10^4$ hr for the industrialization, which is around one year in case the display was powered-on every day. However, low-dimensional perovskite structures are generally more vulnerable to degradation because of their large reactivity based on large surface area. Recently, the operational stability of low-dimensional PeLED was improved to several hours under better stability of the material synthesis and device design. The approach from the material synthesis focused on the A-site cation or ligand engineering [153,154]. Zhao et al. [155] reported the efficient and stable LEDs with $CsPbBr_3$ perovskite quantum dots. They introduced lead naphthenate as lead source for the synthesis. During the synthesis procedure, the naphthenate ion afterward the synthetic reaction roles a ligand to surround perovskite nanocrystal. With these highly stable nanocrystals, they achieved 62 min of LT50. On the other hand, approach from the device design focused on the suppression of ion migration by usage of highly resistive material for the device [156,157]. Liu et al. [158] reported the 216 min of LT50 (time that the device emit 50% of initial luminance) with the device structure of $ITO/NiO_x/perovskite/ZnMgO/Al$. Even though the device was fabricated under all-solution process, the device exhibited long lifetime because the chemically resistive material was used in charge transport layer.

### 4.3. Lasers

Beyond the requirements for perovskite LEDs, application of perovskite lasers is the most challenging subject in perovskite optoelectronic devices. Along with high crystallinity without any defect states and highly dense films, suppressing non-radiative recombination is necessary to achieve population inversion [159–161]. Perovskite materials have demonstrated a high potential for laser applications by exploiting not only slow Auger recombination and slow non-radiative decay but also long carrier diffusion length and a large gain cross-section at the lasing wavelength [7,162]. The realization of perovskite laser applications will enlarge the field of perovskite-based optoelectronic devices for projection-type displays, biomedical imaging, sensors, and various diagnostics (Figure 9a).

### 4.3.1. Key Parameters for Lasers

Although the optical cavity or resonator is essential to provide optical feedback to the gain medium, performance of laser diodes highly depend on the population inversion conditions of the perovskite emitting layer. The population inversion is a prerequisite factor for operating laser diodes, where the majority of electrons are located in an excited state compared to the ground state. In population inversion conditions, which require continuous external pumping (optically or electrically) at the emitting layer, the stimulated emission process exceeds the absorption process. Therefore, since laser diodes require external pumping, lowering the lasing threshold ($\omega_{las}$) has widely been investigated for the commercialization of perovskite-based laser applications.

### 4.3.2. Progess in Perovksite Lasers

G. Xing et al. [4] demonstrated a perovskite laser with $CH_3NH_3PbX_3$ (X = Cl, Br, and I) with a very low lasing threshold. They obtained amplified spontaneous emissions (ASEs) in thin film of a perovskite emission layer in low threshold conditions, where the threshold for the $CH_3NH_3PbI_3/PCBM$ film was measured to be $10 \pm 2$ $\mu J/cm^2$. Moreover, by adopting a mixture of halide X anions ($Cl_3$, $Cl_{1.5}Br_{1.5}$, $Br_3$, $BrI_2$, and $I_3$), they controlled the ASE peak over the entire visible spectral range (from 390 to 790 nm). The successful demonstration of ASEs was obtained by large absorption coefficients, ultralow bulk defect densities and the slow Auger recombination properties of perovskite film. Later, various studies reported the progress of perovskite lasers. Since a nanowire (NW) structure provides both an efficient

gain medium and a cavity, H. Zhu et al. [165] and Y. Fu et al. [28] adopted the perovskite nanowire structure for enhancing the performance of lasers. The single-crystal perovskite nanowires demonstrated a very low lasing threshold (220 nJ/cm$^2$) and high quality factor $Q = \lambda/\delta\lambda \sim 3600$, where $\delta\lambda$ is the full-width-half-maximum (FWHM) of the lasing peak.

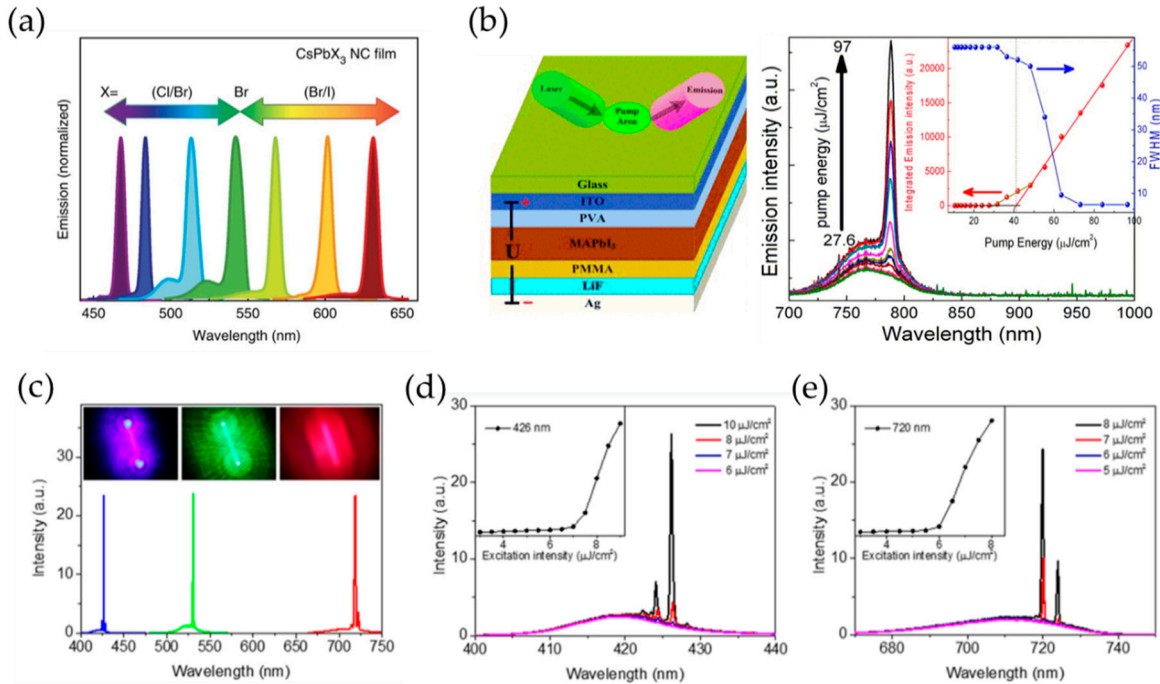

**Figure 9.** (**a**) Mixed halide (i.e., I, Br, and Cl) dependent tunable ASE of CsPbX$_3$ (X = halide) perovskite crystals (pumping intensity range 3–25 μJ/cm$^2$). Adapted with permission from [Low-threshold amplified spontaneous emission and lasing from colloidal nanocrystals of caesium lead halide perovskites]. (**b**) Schematic configuration of the electrically pumped perovskite based laser and ASE of the sample as a function of pumped energy. Adapted with permission from [163]. (**c**) Tunable wavelength of the perovskite crystals by the exchanging the halide. (**d**) Emission spectra of a (**d**) CsPbCl$_3$ and (**e**) CsPbI$_3$ perovskite crystals. Adapted with permission from [164].

Since the operation of a laser requires extremely high operational power conditions, the all-inorganic perovskite NWs have also been investigated. S. W Eaton et al. [166] first demonstrated an all inorganic CsPbBr$_3$ perovskite NW laser. With the delicate design of a single-crystalline CsPbBr$_3$ NW, they achieved a fine lasing threshold of 86 μJ/cm$^2$ along with stable device operation. Later, K. Park et al. [164] and X. Wang et al. [167] demonstrated CsPbX$_3$ perovskite nanowire-based lasers with varying X anions (X = Cl, Br, and I) (Figure 9c,d). The CsPbCl$_3$, CsPbBr$_3$, and CsPbI$_3$ perovskite-based laser showed stimulated emission peaks ranging from 420 nm to 730 nm with low lasing thresholds. Under the optically pumped excitation, exciton-polaritons are formed in the perovskite nanowires which indicates strong exciton-photon coupling in the optical microcavities.

### 4.3.3. Electically Pumped Perovksite Lasers

Accomplishing electrically pumped laser is a long-standing goal in various solution-processed optoelectronics (e.g., Quantum dots and conjugated polymers). Although optically pumped perovskite lasers have achieved very low threshold conditions, maintaining population inversion state through electrical pumping requires much complex and limited conditions. For example, the nanosecond range of biexciton lifetime of perovskite requires high current density up to kA/cm$^2$, which is 2–3 orders magnitude of current density in perovskite-based LEDs. Operation of device in such extremely high biased conditions eventually produces large Joule heating and increase of non-radiative Auger recombination process which cause serious deterioration of device structure and further increase the threshold of devices.

Several attempts have proposed to relieve these challenges. To reduce the Joule heating during high operation condition (~kA/cm$^2$), various heat dissipation strategies have adopted. Balancing charge injection by controlling CTLs, using thermally conductive sapphire substrates, and reducing the pixel sizes are applied to achieve high operational conditions in perovskite based LEDs which have shown EQE of 1% at current density of 1 kA/cm$^2$ [142]. The stable operation at extremely high current density not only suggests the realization of electrically pumped laser, but also enhances the luminescence of perovskite-based LEDs. Similarly, achieving population inversion by applying pulsed voltage is efficient way to reduce heat accumulation where the off state dissipates electrical stress during operations [168]. Integrating a distributed feedback resonator into the perovskite based LEDs structure is also an efficient approach to increase the confinement of a waveguided mode within the perovskite medium, which can reduce threshold condition of devices [169,170].

## 5. Summary and Outlook

The performance of halide perovskite-based optoelectronic devices has been considerably improved through various modifications of perovskite material and device architecture engineering. In this review, we discussed in depth the development of halide perovskites to understand the overall progress in various applications of perovskite materials and to gain insight into the realization of commercialization. The distinct advantageous features of perovskite, such as tunable absorption/emission range, long charge carrier diffusion lengths, and high charge carrier mobility, have made it possible to broaden the application of perovskite-based optoelectronic devices (solar cells, photodetectors, LEDs, and lasers). However, apart from stability issues, several challenges remain to be addressed considering that most of reported progress is still at the early stages of research. Here, we summarized the outlook on future developments of applications as follows.

### 5.1. In Solar Cells

Since the performances of PSCs have exceeded 25% within a very short period of time, there is no doubt that PSCs are one of the promising candidates for commercialization substituting the traditional Si-based solar cell. Considering the growing demand of "affordable and clean energy", which is one of the sustainable development goals (SDGs) defined by the United Nations (UN), the development of new materials for solar energy is highly emphasized for replacing the finite fossil fuels that accelerates natural environment deterioration.

However, there are several remaining factors that have not been clearly unveiled and need to be overcome. Although PSCs show incredible progress, most of the achievement has been done with lead (Pb) halide perovskite materials. Due to the toxicity of Pb, it should be replaced with a nontoxic component to follow the global environmental criteria (e.g., Restricted Hazardous Substances, RoHS). Among various candidates to replace the Pb ions (e.g., Sn, Bi, Sb, and Ge), Sn is widely investigated due to the closest property with Pb. The performance of Sn based PSCs shows steady progress, where the PCE have reached near 10 % within a few decades [171,172]. However, due to the easy oxidation of Sn, lead-free perovskite shows much vulnerable property to air and moisture [173,174]. In addition, the representative solvents for fabricating a highly efficient perovskite layer are toxic solvents (e.g., DMF, DMAc, DMSO, and GBL). Therefore, for real commercialization, it is necessary to replace the fabrication procedure with eco-friendly materials.

### 5.2. In Photodetectors

Photodetectors were one of the most promising applications for organic-inorganic halide perovskite structure. The large absorption coefficient and long diffusion length realizes the high performance of the photodetectors. The most outstanding parameter for the perovskite photodetector is responsivity, which researchers reported for the photodetectors with ~10$^1$ AW$^{-1}$ which is higher than that of the organic photodetector. Moreover, the

perovskite photodetectors exhibit fine characteristics under small external bias voltage, or even zero bias. These characteristics are adequate for the self-powered photodetectors as the key applications for the perovskite photodetectors which detects the photosignal without auxiliary equipment. The self-powered photodetectors can be used only with the simple system and facile fabrication steps that present a wide range of uses, including outdoor environments and mobile applications.

With the common concept that the device converts the photonic energy to electrical energy, perovskite solar cells and perovskite photodetectors share the principal technologies, including the material synthesis and device structures. One of the crucial issues for the high performance especially with perovskite photodetectors is the reduction of dark current, which is the prime factor of the noise current. Many strategies to reduce the dark current and increase the detectivity have been suggested, such as charge barrier layer and crystallinity enhancement.

### 5.3. In Light Emitting Diodes

In light emitting diodes, versatile perovskite structures were studied more actively compared to other applications. The perovskite structures in light emitting diodes are classified by material (MA- or FA-based organic-inorganic hybrid, Cs-based inorganic) and morphological perspectives (three-dimensional structures, quasi-2D sheets, nanowires, nanocrystals, quantum dots, and so on). The perovskite light emitting diodes have advantages in high quantum yield for the photoconversion, wide color range with narrow FWMH, easy color tunability by changing atomic composition. Although there are still several huddles, such as stability and toxicity issues, to be widely accepted as the future display, the perovskite light emitting diodes are establishing steep progress with many studies on both of defect engineering and device designing.

For commercialization of perovskite light emitting diodes, one of the prime challenges is on their stability. The commercial display needs ~$10^4$ h of lifetime but most of recent studies reported only several hours in electroluminescence. To improve the stability of the devices, versatile strategies were suggested, such as A-site cation engineering, ligand engineering, and suppression of ion migration.

### 5.4. In Lasers

The field of perovskite-based lasers is in the early stage of research. By exploiting its great features, the performance of the perovskite laser demonstrates an excellent lasing threshold with a high quality factor. However, most perovskite-based lasing reports are based on optically pumped lasers where achieving electrically driven laser didoes is still an existing challenge (Figure 9b). In particular, the instability of devices at high operational conditions has created various problems, such as Joule heating, increased quenching states, and Auger recombination due to charge injection imbalance. Therefore, reducing these problems is the key to facilitating perovskite as an electrically drivel laser source.

Another challenging issue in perovskite-based lasers is the low performance of blue- or deep blue-emission perovskite materials, where perovskite-based LEDs undergo the same problems. Due to the difficulty in synthesizing an efficient and stable blue-emission perovskite materials, blue perovskite emitter and blue perovskite based LEDs exhibit low PL QYs (<65%) [175,176] and EQE (<6%). To overcome the instability issues in the $Cl^-$ based perovskite structure, Ni ion doping is proposed to obtain violet emission with near-unity PL QYs [177]. Since maintaining material stability is prerequisite factor to achieve ASE emission, these approaches can contribute to the realization of deep blue emission perovskite-based laser applications.

**Author Contributions:** Investigation, S.R. and K.A.; writing—original draft preparation, S.R. and K.A.; writing—review and editing, K.A. and K.-T.K.; supervision, K.A. and K.-T.K.; project administration, K.-T.K.; project administration, K.-T.K.; funding acquisition, K.-T.K. All authors have read and agreed to the published version of the manuscript.

**Funding:** This work was supported by the National Research Foundation of Korea (NRF) grant funded by the Korea government (MSIT) (No. 2020R1A5A1019649 and 2020R1I1A1A01073425).

**Data Availability Statement:** Data sharing not applicable.

**Conflicts of Interest:** The authors declare no conflict of interest. The funders had no role in the design of the study; in the collection, analyses, or interpretation of data; in the writing of the manuscript, or in the decision to publish the results.

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
