# Peer review of "Recent Advances and Challenges in Halide Perovskite Crystals in Optoelectronic Devices from Solar Cells to Other Applications"

_crystals, doi:10.3390/cryst11010039_

Round 1

Reviewer 1 Report

In this manuscript, the developments of various perovskite-based optoelectronic devices determined by different suitable characteristics of halide perovskite materials were discussed in depth. The authors first provided distinct advantageous features of perovskite, and then separately introduced the primary parameters of these devices such as solar cells, photodetectors, LEDs, and lasers. According to the main parameters, the factors affecting the performance of these devices were analyzed.

In order to understand the recent progress of perovskite-based optoelectronic devices as well as the challenges that need to be solved for commercialization, the authors introduced some representative reports on different application fields. The main findings are summarized and the strength of the techniques to overcome the challenges and gain deeper knowledge for further performance improvement is assessed. It is benefit for those who have just entered the research field to establish a macro comprehension of above mentioned.

But this article has some mistakes, and some areas need to be improved:

  1. Equation 4 does not exist in the paper, and here should be expressed by Equation 12.
  2. There is an error occurred in the 114th literature citation.
  3. Figure 8f-g are short of annotations.
  4. It is better to quote more recent reports and make more comparisons.
  5. The interconnection between various fields is also very important, this part should be taken in to account.

Author Response

General Comment: In this manuscript, the developments of various perovskite-based optoelectronic devices determined by different suitable characteristics of halide perovskite materials were discussed in depth. The authors first provided distinct advantageous features of perovskite, and then separately introduced the primary parameters of these devices such as solar cells, photodetectors, LEDs, and lasers. According to the main parameters, the factors affecting the performance of these devices were analyzed.

In order to understand the recent progress of perovskite-based optoelectronic devices as well as the challenges that need to be solved for commercialization, the authors introduced some representative reports on different application fields. The main findings are summarized and the strength of the techniques to overcome the challenges and gain deeper knowledge for further performance improvement is assessed. It is benefit for those who have just entered the research field to establish a macro comprehension of above mentioned.

But this article has some mistakes, and some areas need to be improved:

Response: We sincerely appreciate reviewer #1’s positive evaluation and constructive suggestions on our present work. All authors thoroughly went over comments and suggestions, prepared the responses, and revised the manuscript and the supporting information accordingly. Below, we enclose the point-by-point responses to each comment raised by reviewer #1.

Comment 1: Equation 4 does not exist in the paper, and here should be expressed by Equation 12.

Response: Thank you for your kind comments. We revised the numbering of Equations in our manuscript.

Comment 2: There is an error occurred in the 114th literature citation.

Response: Thank you for your valuable comments. We checked the 113th literature citation was repeated and the repeated citation was eliminated.

Comment 3: Figure 8f-g are short of annotations.

Response: Thank you for your helpful comments. Figure 8 (f) and (g) are schematic diagrams of carrier distributions and recombination rates of active layers by different amount of oleyamine. The figures noted that active layers with small amount of oleyamine showed high carrier mobility. The additional explanation was added in the caption of Figure 8.

Comment 4: It is better to quote more recent reports and make more comparisons.

Response: We appreciate to your comments. We added the more recent reports and more explanation for the perovskite optoelectronics. Not only the academic reports, we also tried to cover more recent result from the news. As a result, about 20 articles were additionally explained in this review paper.

Comment 5: The interconnection between various fields is also very important, this part should be taken in to account.

Response: Thank you for your valuable comments. From the view of energy conversion direction, solar cells and photodetectors have strong interconnection that converted photonic energy to electrical energy. On the other hand, the light emitting diodes and lasers have strong interconnection that converted electrical energy to photonic energy. We added explanation for the interconnection between solar cells and photodetectors in the chapter for photodetector. We also added explanation for the interconnection between light emitting diodes and lasers in the chapter for lasers.

Modifications made in the Manuscript

(Line 762, page 21 in the revised version of Manuscript)

With the common concept that the device converts the photonic energy to the electrical energy, perovskite solar cells and perovskite photodetectors shares the principal technologies including the material synthesis and device structures. Among them, one of the crucial issues for the high performance especially on perovskite photodetectors is at reduction of dark current which is the prime factor of the noise current.

(Line 711, page 19 in the revised version of Manuscript)

Balancing charge injection by controlling CTLs, using thermally conductive sapphire substrates, and reducing the pixel sizes are applied to achieve high operational conditions in perovskite based LEDs which have shown EQE of 1% at current density of 1 kA/cm2. The stable operation at extremely high current density not only suggests the realization of electrically pumped laser but also enhances the luminescence of perovskite based LEDs.

Reviewer 2 Report

 The direction of the manuscript is different from the Crystals journal, and most of the content is known information and general report, and it isn't innovative technology or phenomena, so it is not recommended to publish, i. e., to reject.

Author Response

General Comment: The direction of the manuscript is different from the Crystals journal, and most of the content is known information and general report, and it isn't innovative technology or phenomena, so it is not recommended to publish, i. e., to reject.

Response: We sincerely appreciate reviewer 2’s comments on the present work. We understand the reviewer’s concern about discrepancy between the scope of Crystals journal compared to the subject of present manuscript. The present work much focuses on the various applications of perovskite crystals and partially covering the crystal growth of perovskite materials, which may not match with the main field of the Crystals journal, “understanding of the nucleation, growth, processing, and characterization of crystalline and liquid crystalline materials”. However, the Crystals journal also considers importantly about “mechanical, chemical, electronic, magnetic, and optical properties, and their diverse applications” (Written in the website of Crystals), which exactly matches with present work’s direction. Also, since the present work is prepared for special edition of “Organic inorganic halide perovskite solar cells”, we believe the manuscript is suitable for publication in Crystals.

Reviewer 3 Report

This review paper is quite interesting and useful for the development of electronic devices, optoelectronic devices and Solar cells. But it needs to include more recent results about the solar cells from

(1) The Swiss Center for Electronics and Microtechnology (CSEM) "A 26.5% efficient perovskite-silicon tandem cell Scientists in Switzerland achieved 26.5% efficiency on a perovskite-silicon tandem cell measuring 4cm² and relying on industry-standard screen-printed metallization, further demonstrating the technology’s potential for large-scale production and low-cost electricity generation." and NIMS, Japan too.

(2). I also feel that the LED part is not sufficient and you better to discuss about the current challenges.

(3). Laser part is too short and you better to include some new results if any or at least to you may provide some nice roadmap for future research work/directions of laser devices.

(4). Lastly, you better to include one section about the future challenges and your suggested recommendations for further improvement in the performances of all types of devices. Is there any possibility of Halide
Perovskite Crystals- based UV emitters (LEDs and Laser).

(5). Do not worry for the length of the manuscript and you better to elaborate the research problems in the Halide Perovskite Crystals. You need to mention about the environmental issues related to Halide Perovskite Crystals-based solar cells/LEDs in line to the 17 SDGs of UN.

Please check the English language part carefully.

With this, i would like to recommend this manuscript for major revision.  

Good luck 

The reviewer

Author Response

Reviewer 3

General Comment: This review paper is quite interesting and useful for the development of electronic devices, optoelectronic devices and Solar cells.

Response: We sincerely appreciate reviewer #3’s positive evaluation and constructive suggestions on our present work. All authors thoroughly went over comments and suggestions, prepared the responses, and revised the manuscript and the supporting information accordingly. Below, we enclose the point-by-point responses to each comment raised by reviewer #3.

Comment 1: But it needs to include more recent results about the solar cells from The Swiss Center for Electronics and Microtechnology (CSEM) "A 26.5% efficient perovskite-silicon tandem cell Scientists in Switzerland achieved 26.5% efficiency on a perovskite-silicon tandem cell measuring 4cm² and relying on industry-standard screen-printed metallization, further demonstrating the technology’s potential for large-scale production and low-cost electricity generation." and NIMS, Japan too.

Response: Thank you for your valuable comments. As reviewer commented, the tandem structure of the perovskite solar cells with silicon-based solar cells are one of the main stream for the current perovskite research. CSEM, NIMS, and other research institute achieved high efficiency and high productivity of the solar cells. We added the suggested article and related researches within the revised manuscript.

Modifications made in the Manuscript

(Line 201, page 7 in the revised version of Manuscript)

Industrial level of researches, such as large-scale production, environmental-friendly production, and stable operation, for practical use of PSCs is also investigated widely in various research institutes (e.g. The Swiss Center for Electronics and Microtechnology (CSEM) and National Institute for Materials Science (NIMS)) , which have recently achieved efficiency of 26.5% in perovskite/Si multijunction solar cells in large-area production. 

Comment 2: I also feel that the LED part is not sufficient and you better to discuss about the current challenges.

Response: Thank you for your helpful comments. One of the prime challenges in PeLEDs is stability issue. Though the PeLEDs achieved outstanding improvement in whole of parameters by many researchers, the lifetime of the device was still too short to commercialize it. We added the explanation for the stability issue as the challenges in PeLEDs and strategies to increase the lifetime.

Modifications made in the Manuscript

(Line 630, page 17 in the revised version of Manuscript)

Despite of these extraordinary characteristics of the perovskite light emitting diodes, the commercialization of these device are still challenging due to the insufficient stability. Not only the light emitting devices, all the optoelectronic devices with perovskite structures are facing stability issue but low-dimensional perovskite structures for light emitting diodes are having more difficulty on it. The display needs long lifetime over 104 hr for the industrialization which is around 1 year in case the display was powered-on every day. However, low-dimensional perovskite structures are generally more vulnerable to degradation because of their large reactivity based on large surface area. Recently, the operational stability of low-dimensional PeLED was improved to several hours under better stability of the material synthesis and device design. The approach from the material synthesis focused on the A-site cation or ligand engineering . Zhao et al. reported  the efficient and stable LEDs with CsPbBr3 perovskite quantum dots. They introduced lead naphthenate as lead source for the synthesis. During the synthesis procedure, the naphthenate ion afterward the synthetic reaction roles a ligand to surround perovskite nanocrystal. With these highly stable nanocrystals, they achieved 62 min of LT50. On the other hand, approach from the device design focused on the suppression of ion migration by usage of highly resistive material for the device . Liu et al. reported  the 216 min of LT50 (time that the device emit 50% of initial luminance) with the device structure of ITO / NiOx / perovskite / ZnMgO / Al. Even though the device was fabricated under all-solution process, the device exhibited long lifetime because the chemically resistive material was used in charge transport layer.

Comment 3: Laser part is too short and you better to include some new results if any or at least to you may provide some nice roadmap for future research work/directions of laser devices.

Response: We appreciate to your comments. Since the perovskite-based lasers is in the early stage of research, we briefly address the progress of optically pumping type of perovskite based lasers. As reviewer commented, we have additionally addressed the undergoing challenges to achieve electrically pumped lasing in perovskite based lasers and several meaningful approaches to solve these issues within the revised manuscript.

Modifications made in the Manuscript

(Line 700, page 19 in the revised version of Manuscript)

4.3.3. Electically Pumped Perovksite Lasers

Accomplishing electrically pumped laser is a long-standing goal in various solution-processed optoelectronics (e.g. Quantum dots and conjugated polymers). Although optically pumped perovskite lasers have achieved very low threshold conditions, maintaining population inversion state through electrical pumping requires much complex and limited conditions. For example, the nanosecond range of biexciton lifetime of perovskite requires high current density up to kA/cm2, which is 2-3 orders magnitude of current density in perovskite-based LEDs. Operation of device in such extremely high biased conditions eventually produces large Joule heating and increase of non-radiative Auger recombination process which cause serious deterioration of device structure and further increase the threshold of devices.

Several attempts have proposed to relieve these challenges. To reduce the Joule heating during high operation condition (~kA/cm2), various heat dissipation strategies have adopted. Balancing charge injection by controlling CTLs, using thermally conductive sapphire substrates, and reducing the pixel sizes are applied to achieve high operational conditions in perovskite based LEDs which have shown EQE of 1% at current density of 1 kA/cm2. The stable operation at extremely high current density not only suggests the realization of electrically pumped laser but also enhances the luminescence of perovskite based LEDs. Similarly, achieving population inversion by applying pulsed voltage is efficient way to reduce heat accumulation where the off state dissipates electrical stress during operations. Integrating distributed feedback resonator into perovskite based LEDs structure is also efficient approach to increase the confinement of a waveguided mode within the perovskite medium, which can reduce threshold condition of devices.

Comment 4: Lastly, you better to include one section about the future challenges and your suggested recommendations for further improvement in the performances of all types of devices. Is there any possibility of Halide Perovskite Crystals- based UV emitters (LEDs and Laser).

Response: Thank you for your valuable comments. According to the atomic composition of perovskite structure, the emission spectrum can be tuned in a wide wavelength range from UV to NIR. These characteristics means that halide perovskite UV emitter is theoretically possible. Nevertheless, realization of perovskite UV emitter is facing several issues which is similar to those of blue and deep blue emitter. We have added future challenges of each device applications within the revised manuscript. Since the performance of blue- and deep blue- emission perovskite based LEDs and Laser applications is lack behind, we have focused on their issues.

Modifications made in the Manuscript

(Line 487, page 14 in the revised version of Manuscript)

Kumawat et al. reported the bandgap tuning of CH3NH3Pb(Br1-xClx)3 perovskite structure by varying x from 0 to 1. The bandgap of the perovskite film was tuned from 2.42eV to 3.16eV that covered ultraviolet (409nm) to visible light. Eperon et al. controlled the bandgap of FAPbIyBr3-y perovskite structure by varying y from 0 to 1 which covered from visible to near infrared (NIR) (840nm). These wavelength tunability of perovskite structure under compositional control enables the color controllability in the full range of visible light, which creates the key advantages in full color or white displays, lighting applications, and so on. Also, wide tunable perovskite structures can expand their applications to UV LEDs and NIR LEDs.

(Line 744, page 20 in the revised version of Manuscript)

Among various candidates to replace the Pb ions (e.g. Sn, Bi, Sb, and Ge), Sn is widely investigated due to the closest property with Pb. The performance of Sn based PSCs shows steady progress, where the PCE have reached near 10 % within a few decades. However, due to the easy oxidation of Sn, lead-free perovskite shows much vulnerable property to air and moisture.

(Line 792, page 20 in the revised version of Manuscript)

Another challenging issue in perovskite based lasers is low performance of blue- or deep blue- emission perovskite materials, where perovskite based LEDs undergoes same problems. Due to the difficulty in synthesizing an efficient and stable blue-emission perovskite materials, blue perovskite emitter and blue perovskite based LEDs exhibit low PL QYs ( < 65 %) and EQE ( < 6 %). To overcome the instability issues in Cl- based perovskite structure, Ni ion doping is proposed to obtain violet emission with near-unity PL QYs. Since maintaining material stability is prerequisite factor to achieve ASE emission, the approaches can contribute to realization of deep blue emission perovskite based laser applications.

Comment 5: Do not worry for the length of the manuscript and you better to elaborate the research problems in the Halide Perovskite Crystals. You need to mention about the environmental issues related to Halide Perovskite Crystals-based solar cells/LEDs in line to the 17 SDGs of UN.

Response: We have added the importance of environmental-friendly renewable energy based on the sustainable development goals (SDGs) defined by the United Nations (UN) within the revised manuscript.

Modifications made in the Manuscript

(Line 194, page 7 in the revised version of Manuscript)

With continuously increasing population along with high criteria for future renewable energy targets, there is intensive needs for environmental-friendly energies such as solar cells.

(Line 736, page 19 in the revised version of Manuscript)

Considering the growing demand of “affordable and clean energy”, which is one of the sustainable development goals (SDGs) defined by the United Nations (UN), the development of new materials for solar energy is highly emphasized for replacing the finite fossil fuels that accelerates natural environment deterioration.

Round 2

Reviewer 2 Report

Consortium blockchain is still private to a small set of nodes. Though I still couldn't convince myself it's a worthy publication for Crystals, I think publishing it won't be disastrous for Crystals.

Reviewer 3 Report

Dear Authors

Great job. It seems like a real review paper. However you better to update the most recent world record achievement by Oxford PV, where perovskite-silicon tandem solar cells gives PCE 29.52 %. Crossed to the HZB record. Please also check plagiarism test via Crosscheck. With this I would like to accept your manuscript after minor addition of Oxford PV updates.

Cheers 

Good luck 

The reviewer